# QUATERNION EQUIVARIANT CAPSULE NETWORKS FOR 3D POINT CLOUDS

## ABSTRACT

We present a 3D capsule architecture for processing of point clouds that is equivariant with respect to the $SO(3)$ rotation group, translation and permutation of the unordered input sets. The network operates on a sparse set of local reference frames, computed from an input point cloud and establishes end-to-end equivariance through a novel 3D quaternion group capsule layer, including an equivariant dynamic routing procedure. The capsule layer enables us to disentangle geometry from pose, paving the way for more informative descriptions and a structured latent space. In the process, we theoretically connect the process of dynamic routing between capsules to the well-known Weiszfeld algorithm, a scheme for solving *iterative re-weighted least squares (IRLS)* problems with provable convergence properties, enabling robust pose estimation between capsule layers. Due to the sparse equivariant quaternion capsules, our architecture allows joint object classification and orientation estimation, which we validate empirically on common benchmark datasets.

## 1 INTRODUCTION

It is now well understood that in order to learn a compact representation of the input data, one needs to respect the symmetries in the problem domain (Cohen et al., 2019; Weiler et al., 2018a). Arguably, one of the primary reasons of the success of 2D convolutional neural networks (CNN) is the *translation-invariance* of the 2D convolution acting on the image grid (Giles & Maxwell, 1987; Kondor et al., 2018). Recent trends aim to transfer this success into the 3D domain in order to support many applications such as shape retrieval, shape manipulation, pose estimation, 3D object modeling and detection. There, the data is naturally represented as sets of 3D points or a *point cloud* (Qi et al., 2017a;b). Unfortunately, extension of CNN architectures to 3D point clouds is non-trivial due to two reasons: 1) point clouds are irregular and unstructured, 2) the group of transformations that we are interested in is more complex as 3D data is often observed under arbitrary non-commutative $SO(3)$ rotations. As a result, achieving appropriate embeddings requires 3D networks that work on points to be *equivariant* to these transformations, while also being invariant to the permutations of the point set.

In order to fill this important gap, we propose the quaternion equivariant point capsule network or *QE-Network* that is suited to process point clouds and is equivariant to $SO(3)$ rotations compactly parameterized by quaternions (Fig. 2), in addition to preserved translation and permutation equivariance. Inspired by the local group equivariance (Lenssen et al., 2018; Cohen et al., 2019), we efficiently cover $SO(3)$ by restricting ourselves to the sparse set of local reference frames (LRF) that collectively characterize the object orientation. The proposed capsule layers (Hinton et al., 2011) deduces equivariant latent representations by robustly combining those local LRFs using the proposed *Weiszfeld dynamic routing*. Hence, our latent features specify to local orientations disentangling the pose from object existence. Such explicit storage is unique to our work and allows us to perform rotation estimation jointly with object classification. Our final architecture is a hierarchy of QE-networks, where we use classification error as the only training cue and adapt a Siamese version when the relative rotation is to be regressed. We neither explicitly supervise the network with pose annotations nor train by augmenting rotations. Overall, our contributions are:

1. We propose a novel, fully $SO(3)$-equivariant capsule architecture that is tailored for simultaneous classification and pose estimation of 3D point clouds. This network produces in-

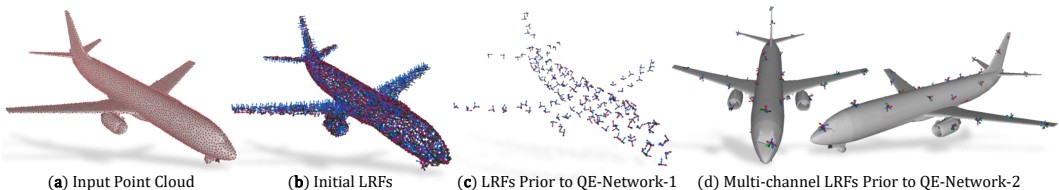

(**a**) Input Point Cloud     (**b**) Initial LRFs     (**c**) LRFs Prior to QE-Network-1     (**d**) Multi-channel LRFs Prior to QE-Network-2

Figure 1: Our network operates by processing local reference frames (LRF) on the object. Initial LRFs (**b**) are obtained by computing normal & tangent vectors on the point set in (**a**). (**c**) shows the LRFs randomly sampled from (a) and these are inputs to the first layer of our network. Subsequently, we obtain a multi-channel LRF that is a set of reference frames per pooling center (**d**). Holistically, our network aggregates the LRFs to arrive at rotation equivariant capsules.

variant latent representations while explicitly decoupling the orientation into capsules, thus attaining equivariance. Note that equivariance results have not been previously achieved regarding the quaternion parameterization of the 3D special orthogonal group.

2. By utilizing LRFs on points, we reduce the space of orientations that we consider and hence can work sparsely on a subset of the group elements.

3. We theoretically prove the equivariance properties of our 3D network regarding the quaternion group. Moreover, to the best of our knowledge, we for the first time establish a connection between the dynamic routing of Sabour et al. (2017) and Generalized Weiszfeld iterations (Aftab et al., 2015). By that, we theoretically argue for the convergence of the employed dynamic routing.

4. We experimentally demonstrate the capabilities of our network on classification and orientation estimation of 3D shapes.

## 2 PRELIMINARIES AND TECHNICAL BACKGROUND

In this paper we will speak of the equivariance of point clouds under the actions of quaternions. We now provide the necessary background required for the grasp of this content.

### 2.1 EQUIVARIANCE

**Definition 1** (Equivariant Map). *For a $\mathcal{G}$-space acting on $\mathcal{X}$, the map $\Phi : \mathcal{G} \times \mathcal{X} \mapsto \mathcal{X}$ is said to be equivariant if its domain and co-domain are acted on by the same symmetry group (Cohen & Welling, 2016; Cohen et al., 2018a):*

$$\Phi(\mathbf{g}_1 \circ \mathbf{x}) = \mathbf{g}_2 \circ \Phi(\mathbf{x}) \tag{1}$$

*where $\mathbf{g}_1 \in \mathcal{G}$ and $\mathbf{g}_2 \in \mathcal{G}$. Equivalently $\Phi(T(\mathbf{g}_1)\,\mathbf{x}) = T(\mathbf{g}_2)\,\Phi(\mathbf{x})$, where $T(\cdot)$ is a linear representation of the group $\mathcal{G}$. Note that $T(\cdot)$ does not have to commute. It suffices for $T(\cdot)$ to be a homomorphism: $T(\mathbf{g}_1 \circ \mathbf{g}_2) = T(\mathbf{g}_1) \circ T(\mathbf{g}_2)$. In this paper we use a stricter form of equivariance and consider $\mathbf{g}_2 = \mathbf{g}_1$.*

**Definition 2** (Equivariant Network). *An architecture or network is said to be equivariant if all of its layers are equivariant maps. Due to the transitivity of the equivariance, stacking up equivariant layers will result in globally equivariant networks e.g., rotating the input will produce output vectors which are transformed by the same rotation (Lenssen et al., 2018; Kondor & Trivedi, 2018).*

### 2.2 THE QUATERNION GROUP $\mathbb{H}_1$

The choice of 4-vector quaternions has multiple motivations: 1. All 3-vector formulations suffer from infinitely many singularities as angle goes to $0$, whereas quaternions avoid those. 2. 3-vectors also suffer from infinitely many redundancies (the norm can grow indefinitely). Quaternions have a single redundancy: $q = -q$, a condition that is in practice easy to enforce. 3. Computing the actual 'manifold mean' on the Lie algebra requires iterative techniques with subsequent updates on the tangent space. Such iterations are computationally harmful for a differentiable GPU implementation.

**Definition 3** (Quaternion). *A quaternion $\mathbf{q}$ is an element of Hamilton algebra $\mathbb{H}_1$, extending the complex numbers with three imaginary units $\boldsymbol{i}, \boldsymbol{j}, \boldsymbol{k}$ in the form: $\mathbf{q} = q_1\boldsymbol{1} + q_2\boldsymbol{i} + q_3\boldsymbol{j} + q_4\boldsymbol{k} =$*

---

**Algorithm 1:** Quaternion Equivariant Dynamic Routing

---

1  **input**  : Input points $\{\mathbf{x}_1, ..., \mathbf{x}_K\} \in \mathbb{R}^{K \times 3}$, input capsules (LRFs) $\mathcal{Q} = \{\mathbf{q}_1, \ldots, \mathbf{q}_L\} \in \mathbb{H}_1{}^L$,
       with $L = N^c \cdot K$, $N^c$ is the number of capsules per point, activations $\boldsymbol{\alpha} = (\alpha_1, \ldots, \alpha_L)^T$,
       trainable transformations $\mathcal{T} = \{\mathbf{t}_{i,j}\}_{i,j} \in \mathbb{H}_1{}^{L \times M}$

2  **output:** Updated frames $\hat{\mathcal{Q}} = \{\hat{\mathbf{q}}_1, \ldots, \hat{\mathbf{q}}_M\} \in \mathbb{H}_1{}^M$, updated activations $\hat{\boldsymbol{\alpha}} = (\hat{\alpha}_1, \ldots, \hat{\alpha}_M)^T$

3  **for** *All primary (input) capsules $i$* **do**

4      **for** *All latent (output) capsules $j$* **do**

5         $\mathbf{v}_{i,j} \leftarrow \mathbf{q}_i \circ \mathbf{t}_{i,j}$  `// compute votes`

6  **for** *All latent (output) capsules $j$* **do**

7      $\hat{\mathbf{q}}_j \leftarrow \mathcal{A}\big(\{\mathbf{v}_{1,j} \ldots \mathbf{v}_{K,j}\}, \boldsymbol{\alpha}\big)$  `// initialize output capsules`

8      **for** *$k$ iterations* **do**

9         **for** *All primary (input) capsules $i$* **do**

10           $w_{i,j} \leftarrow \alpha_i \cdot \text{sigmoid}\big(-\delta(\hat{\mathbf{q}}_j, \mathbf{v}_{i,j})\big)$  `// compute the current weight`

11         $\hat{\mathbf{q}}_j \leftarrow \mathcal{A}\big(\{\mathbf{v}_{1,j} \ldots \mathbf{v}_{L,j}\}, \mathbf{w}_{:,j}\big)$  `// see Eq (4)`

12      $\hat{\alpha}_j \leftarrow \text{sigmoid}\big(-\frac{1}{K} \sum_1^L \delta(\hat{\mathbf{q}}_j, \mathbf{v}_{i,j})\big)$ `// recompute activations`

---

$(q_1, q_2, q_3, q_4)^T$, *with* $(q_1, q_2, q_3, q_4)^T \in \mathbb{R}^4$ *and* $\boldsymbol{i}^2 = \boldsymbol{j}^2 = \boldsymbol{k}^2 = \boldsymbol{ijk} = -\boldsymbol{1}$. $q_1 \in \mathbb{R}$ *denotes the scalar part and* $\boldsymbol{v} = (q_2, q_3, q_4)^T \in \mathbb{R}^3$, *the vector part. The* conjugate $\bar{\mathbf{q}}$ *of the quaternion* $\mathbf{q}$ *is given by* $\bar{\mathbf{q}} := q_1 - q_2\boldsymbol{i} - q_3\boldsymbol{j} - q_4\boldsymbol{k}$. *A* unit quaternion $\mathbf{q} \in \mathbb{H}_1$ *with* $1 \stackrel{!}{=} \|\mathbf{q}\| := \mathbf{q} \cdot \bar{\mathbf{q}}$ *and* $\mathbf{q}^{-1} = \bar{\mathbf{q}}$, *gives a compact and numerically stable parametrization to represent orientation of objects on the unit sphere* $\mathcal{S}^3$, *avoiding gimbal lock and singularities (Busam et al., 2017). Identifying antipodal points* $\mathbf{q}$ *and* $-\mathbf{q}$ *with the same element, the unit quaternions form a double covering group of* $SO\,(3)$. $\mathbb{H}_1$ *is closed under the non-commutative multiplication or the Hamilton product:*

$$(\mathbf{p} \in \mathbb{H}_1) \circ (\boldsymbol{r} \in \mathbb{H}_1) = [p_1 r_1 - \mathbf{v}_p \cdot \mathbf{v}_r \,;\, p_1 \mathbf{v}_r + r_1 \mathbf{v}_p + \mathbf{v}_p \times \mathbf{v}_r]. \tag{2}$$

**Definition 4** (Linear Representation of $\mathbb{H}_1$). *We follow Birdal et al. (2018) and use the*

$$\mathbf{T}(\mathbf{q}) \triangleq \begin{bmatrix} q_1 & -q_2 & -q_3 & -q_4 \\ q_2 & q_1 & -q_4 & q_3 \\ q_3 & q_4 & q_1 & -q_2 \\ q_4 & -q_3 & q_2 & q_1 \end{bmatrix}.$$

*To be concise we will use capital letters to refer to the matrix representation of quaternions e.g.* $\mathbf{Q} \equiv T(\mathbf{q})$, $\mathbf{G} \equiv T(\mathbf{g})$. *Note that* $T(\cdot)$, *the injective homomorphism to the orthonormal matrix ring, by construction satisfies the condition in Dfn. 1 (Steenrod, 1951):* $\det(\mathbf{Q}) = 1, \mathbf{Q}^\top = \mathbf{Q}^{-1}, \|\mathbf{Q}\| = \|\mathbf{Q}_{i,:}\| = \|\mathbf{Q}_{:,i}\| = 1$ *and* $\mathbf{Q} - q_1\mathbf{I}$ *is skew symmetric:* $\mathbf{Q} + \mathbf{Q}^\top = 2q_1\mathbf{I}$. *It is easy to verify these properties. $T$ linearizes the Hamilton product or the group composition:* $\mathbf{g} \circ \mathbf{q} \triangleq T(\mathbf{g})\mathbf{q} \triangleq \mathbf{G}\mathbf{q}$.

## 2.3  3D Point Clouds

**Definition 5** (Point Cloud). *We define a 3D surface to be a differentiable 2-manifold embedded in the ambient 3D Euclidean space:* $\mathcal{M}^2 \in \mathbb{R}^3$ *and a point cloud to be a discrete subset sampled on* $\mathcal{M}^2$: $\mathbf{X} \in \{\mathbf{x}_i \in \mathcal{M}^2 \cap \mathbb{R}^3\}$.

**Definition 6** (Local Geometry). *For a smooth point cloud* $\{\mathbf{x}_i\} \in \mathcal{M}^2 \subset \mathbb{R}^{N \times 3}$, *a* local reference frame *(LRF) is defined as an ordered basis of the tangent space at* $\mathbf{x}$, $\mathcal{T}_{\mathbf{x}}\mathcal{M}$, *consisting of orthonormal vectors:* $\mathcal{L}(\mathbf{x}) = [\boldsymbol{\partial}_1, \boldsymbol{\partial}_2, \boldsymbol{\partial}_3 \equiv \boldsymbol{\partial}_1 \times \boldsymbol{\partial}_2]$. *Usually the first component is defined to be the surface normal* $\boldsymbol{\partial}_1 \triangleq \mathbf{n} \in \mathcal{S}^2 : \|\mathbf{n}\| = 1$ *and the second one is picked according to a modality dependent heuristic.*

Note that recent trends such as (Cohen et al., 2019) acknowledge the ambiguity and either employ a *gauge* (tangent frame) equivariant design or propagate the determination of a certain direction until the last layer (Poulenard & Ovsjanikov, 2018). Here, we will assume that $\boldsymbol{\partial}_2$ can be uniquely and

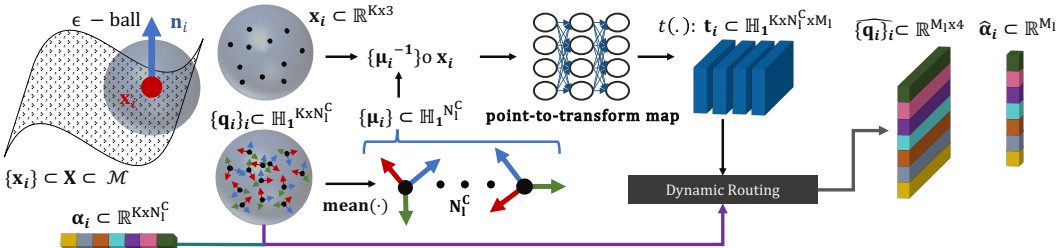

Figure 2: Our **q**uaternion **e**quivariant (QE) network for processing local patches: Our input is a 3D point set $\mathbf{X}$ on which we query local neighborhoods $\{\mathbf{x}_i\}$ with precomputed LRFs $\{\mathbf{q}_i\}$. Essentially, we learn the parameters of a fully connected network that continuously maps the canonicalized local point set to transformations $\mathbf{t}_i$, which are used to compute hypotheses (votes) from input poses. By a special dynamic routing procedure that uses the activations determined in a previous layer, we arrive at latent capsules that are composed of a set of orientations $\hat{\mathbf{q}}_i$ and new activations $\hat{\boldsymbol{\alpha}}_i$. Thanks to the decoupling of local reference frames, $\hat{\boldsymbol{\alpha}}_i$ is invariant and orientations $\hat{\mathbf{q}}_i$ are equivariant to input rotations. All the operations and hence the entire QE-network are equivariant achieving a guaranteed disentanglement of the rotation parameters. *Hat symbol* ($\hat{\mathbf{q}}$) *refers to 'estimated'.*

repeatably computed, a reasonable assumption for the point sets we consider (Petrelli & Di Stefano, 2011). For the cases where this does not hold, we will rely on the network's robustness. We will explain our method of choice in Sec. 4 and visualize LRFs of an airplane object in Fig. 1.

## 3 $SO(3)$-EQUIVARIANT 3D CAPSULE NETWORKS

Disentangling orientation from representations requires guaranteed equivariances and invariances. Yet, the original capsule networks of Sabour et al. (2017) cannot achieve equivariance to general groups. To this end, Lenssen et al. (2018) proposed to use a manifold-mean and a special aggregation that makes sure that the trainable transformations get pose-aligned points as input. We will extend this idea to the non-abelian $SO(3)$ and design capsule networks sparsely operating on a set of LRFs computed on local neighborhoods of points, parameterized by quaternions. In the following, we first explain our novel capsule layers, the main building block of our architecture. We then show how to stack those layers via a simple aggregation resulting in an $SO(3)$-equivariant 3D capsule network that yields invariant representations (or activations) as well as equivariant rotations (latent capsules).

### 3.1 QUATERNION EQUIVARIANT CAPSULE LAYERS

To construct equivariant layers on the group of rotations, we are required to define a left-equivariant averaging operator $\mathcal{A}$ that is invariant under permutations of the group elements, as well as a distance metric $\delta$ that remains unchanged under the action of the group. For these, we make the following choices:

**Definition 7** (Geodesic Distance). *The Riemannian (geodesic) distance in the manifold of rotations lead to the following geodesic distance $\delta(\cdot) \equiv d_{quat}(\cdot)$:*

$$d(\mathbf{q}_1, \mathbf{q}_2) \equiv d_{quat}(\mathbf{q}_1, \mathbf{q}_2) = 2\cos^{-1}(|\langle \mathbf{q}_1, \mathbf{q}_2 \rangle|) \qquad (3)$$

**Definition 8** (Quaternion Mean $\boldsymbol{\mu}(\cdot)$). *For a set of $Q$ rotations $\mathbf{S} = \{\mathbf{q}_i\}$ and associated weights $\mathbf{w} = \{w_i\}$, the weighted mean operator $\mathcal{A}(\mathbf{S}, \mathbf{w}) : \mathbb{H}_1^n \times \mathbb{R}^n \mapsto \mathbb{H}_1^n$ is defined through the following maximization procedure (Markley et al., 2007):*

$$\bar{\mathbf{q}} = \underset{\mathbf{q} \in \mathbb{S}^3}{\arg\max}\ \mathbf{q}^\top \mathbf{M} \mathbf{q} \qquad (4)$$

*where $\mathbf{M} \in \mathbb{R}^{4 \times 4}$ is defined as: $\mathbf{M} \triangleq \sum\limits_{i=1}^{Q} w_i \mathbf{q}_i \mathbf{q}_i^\top$. The average quaternion is the eigenvector of $\mathbf{M}$ corresponding to the maximum eigenvalue. This operation lends itself to both analytic (Magnus, 1985) and automatic differentiation (Laue et al., 2018).*

**Theorem 1.** *Quaternions, the employed mean $\mathcal{A}(\mathbf{S}, \mathbf{w})$ and geodesic distance $\delta(\cdot)$ enjoy the following properties:*

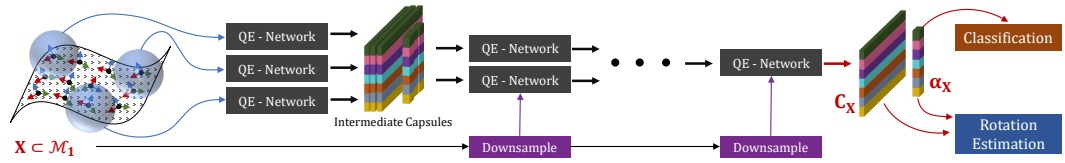

Figure 3: Our entire capsule architecture. We hierarchically send all the local patches to our Q-network as shown in Fig. 2. At each level the points are pooled in order to increase the receptive field, gradually reducing the LRFs into a single capsule per class. We use classification and pose estimation (in the siamese case) as supervision cues to train the point-to-transform maps.

1. $\mathcal{A}(\mathbf{g} \circ \mathbf{S}, \mathbf{w})$ *is left-equivariant:* $\mathcal{A}(\mathbf{g} \circ \mathbf{S}, \mathbf{w}) = \mathbf{g} \circ \mathcal{A}(\mathbf{S}, \mathbf{w})$.

2. *Operator* $\mathcal{A}$ *is invariant under permutations:* $\mathcal{A}(\{\mathbf{q}_{\sigma(1)}, \dots, \mathbf{q}_{\sigma(Q)}\}, \mathbf{w}_\sigma) = \mathcal{A}(\{\mathbf{q}_1, \dots, \mathbf{q}_Q\}, \mathbf{w})$.

3. *The transformations* $\mathbf{g} \in \mathbb{H}_1$ *preserve the geodesic distance* $\delta(\cdot)$ *given in Dfn. 7.*

*Proof.* The proofs are given in the supplementary material. $\square$

We also note that the above mean is closed form, differentiable and can be implemented batchwise. We are now ready to construct the group dynamic routing (DR) by agreement that is equivariant thanks to Thm. 1. The core idea is to *route* from or assign the *primary capsules* that constitute the input LRF set, to the *latent capsules* by an iterative clustering which respects the group structure. At each step, we assign the weighted group mean to each output capsule. The weights $w \leftarrow \sigma(\mathbf{x}, \mathbf{y})$ are inversely propotional to the distance between the vote quaternion and the new quaternion (cluster center). See Alg. 1 for details. In the following, we analyze our variant of routing as an interesting case of the affine, Riemannian Weiszfeld algorithm (Aftab et al., 2015; 2014).

**Lemma 1.** *For* $\sigma(\mathbf{x}, \mathbf{y}) = \delta(\mathbf{x}, \mathbf{y})^{q-2}$ *the equivariant routing procedure given in Alg. 1 is a variant of the affine subspace Wieszfeld algorithm (Aftab et al., 2015; 2014) that is a robust algorithm for computing the* $L_q$ *geometric median.*

*Proof Sketch.* The proof follows from the definition of Weiszfeld iteration (Aftab et al., 2014) and the mean and distance operators defined in Sec. 3.1. We first show that computing the weighted mean is equivalent to solving the normal equations in the iteratively reweighted least squares (IRLS) scheme (Burrus, 2012). Then, the inner-most loop correspond to the IRLS or Weiszfeld iterations. We provide the detailed proof in supplementary material. $\square$

Note that, in practice one is quite free to choose the weighting function $\sigma(\cdot)$ as long as it is inversely proportional to the geodesic distance and concave (Aftab & Hartley, 2015). We leave the analyses of the variants of these algorithms as a future work. The original dynamic routing can also be formulated as a clustering procedure with a KL divergence regularization. This holistic view paves the way to better routing algorithms (Wang & Liu, 2018). Our perspective is akin yet more geometric due to the group structure of the parameter space. Thanks to the connection to Weiszfeld algorithm, the convergence behavior of our dynamic routing can be directly analyzed within the theoretical framework presented by Aftab et al. (2014; 2015).

**Theorem 2.** *Under mild assumptions provided in the appendix, the sequence of the DR-iterates generated by the inner-most loop almost surely converges to a critical point.*

*Proof Sketch.* Proof, given in the appendix, is a direct consequence of Lemma 1 and directly exploits the connection to the Weiszfeld algorithm. $\square$

### 3.2 EQUIVARIANT 3D POINT CAPSULE NETWORK ARCHITECTURE

The essential ingredient of our architecture, QE-Network, is shown in Fig. 2. We also provide a corresponding pseudocode in Alg. 3 of suppl. material. The input of the QE-Network are a local patch of points with coordinates $\mathbf{x}_i \subset \mathbb{R}^{K \times 3}$, rotations parametrized as quaternions $\mathbf{q}_i \subset \mathbb{H}_1^{K \times N^c}$ and activations $\boldsymbol{\alpha}_i \subset \mathbb{R}^{K \times N^c}$. $\mathbf{q}_i$ also represents input primary capsules and local reference frames. $N^c$ is the number of input capsule channels per point and it is equal to the number of output capsules

Table 1: Classification accuracy on ModelNet40 dataset (Wu et al., 2015) for different methods as well as ours. We also report the number of parameters optimized for each method. **X/Y** means that we train with **X** and test with **Y**.

|  | PN | PN++ | KD-treeNet | Point2Seq | Sph.CNNs | PRIN | PPF | Ours (Var.) | Ours |
|---|---|---|---|---|---|---|---|---|---|
| **NR/NR** | 88.45 | 89.82 | 86.20 | **92.60** | - | 80.13 | 70.16 | 85.27 | 74.43 |
| **NR/AR** | 12.47 | 21.35 | 8.49 | 10.53 | 43.92 | 68.85 | 70.16 | 11.75 | **74.07** |
| **# Params** | 3.5M | 1.5M | 3.6M | 1.8M | 0.5M | 1.5M | 3.5M | 0.4M | **0.4M** |

($M$) from the last layer. In the initial layer, $\mathbf{q}_i$ represents the pre-computed LRFs and $N^c$ is equal to 1. Given points $\mathbf{x}_i$ and rotations $\mathbf{q}_i$, we compute the quaternion average $\mu_i$ in channel-wise as the initial pose candidates: $\mu_i \subset \mathbb{H}_1{}^{N^c}$. These candidates are used to bring the receptive field in multiple canonical orientations by rotating the points: $\mathbf{x}'_i = (\mu_i{}^{-1} \circ \mathbf{x}_i) \subset \mathbb{R}^{K \times N^c \times 3}$. Since the points in the local receptive field lie in continuous $\mathbb{R}^3$, training a discrete set of pose transformations $\mathbf{t}_{i,j}$ based on local coordinates is not possible. Instead, we employ a point-to-transform network $t(\cdot) : \mathbb{R}^{N^c \times 3} \to \mathbb{R}^{M \times N^c \times 4}$ that maps the point in multiple canonical orientations to transformations. The network is shared over all points to compute the transformations $\mathbf{t}_{i,j} = (t(\mathbf{x}'_1), ..., t(\mathbf{x}'_K))_{i,j} \subset \mathbb{R}^{K \times M \times N^c \times 4}$, which are used to calculate the votes for dynamic routing as $\mathbf{v}_{i,j} = \mathbf{q}_i \circ \mathbf{t}_{i,j}$. The network $t(\cdot)$ consists of fully-connected layers that regresses the transformations, similar to common operators for continuous convolutions (Schütt et al., 2017; Wang et al., 2018; Fey et al., 2018). It is the continuous alternative to directly optimizing transformations lying in a grid kernel, as it is done in the original dynamic routing by Sabour et al. (2017). Note that $t(\cdot)$ predicts quaternions by unit-normalizing the regressed output: $\mathbf{t}_{i,j} \subset \mathbb{H}_1{}^{K \times M \times N^c}$. Although Riemannian layers of Bécigneul & Ganea (2018) or spherical predictions of Liao et al. (2019) can improve the performance, the simple strategy works reasonably for our case. After computing the votes, we utilize the input activation $\boldsymbol{\alpha}_i$ to iteratively refine the output capsules (weighted average of votes) $\hat{\mathbf{q}}_i$ and activations $\hat{\boldsymbol{\alpha}}_i$ by routing by agreement as shown in Alg. 1.

In order to gradually increase the receptive field, we stack QE-networks creating a deep hierarchy, pooling the points and the LRFs before each layer. Note that we are allowed to do so thanks to the properties of equivariance. In particular, we input $N = 64$ patches to our architecture that is composed of two QE-networks. We call the centers of these patches *pooling centers*. In the first layer, each of those centers is linked to their immediate vicinity leading to $K = 9$-star local connectivity from which we compute the $64 \times 64 \times 4$ intermediary capsules. The input LRFs of the first layer are sampled from pre-calculated LRF-set and the input activation is set to 1. The LRFs in the second layer $l = 2$ are the output capsules of the first layer, $l = 1$ and are routed to the output capsules that are as many as the number of classes $C$, $M_2 = C$. The activation of the second layer is updated by the output of the first layer as well. This construction is shown in Fig. 3. Specifically, for $l = 1$, we use $K = 9, N_l{}^c = 1, M_l = 64$ and for $l = 2$, $K = 64, N_l{}^c = 64, M_l = C = 40$. This way, in this last layer all the pooling centers act as a single patch ($K = 64$). A single QE-network acts on this patch to create the final $C \times 4$ capsules and $C$ activations. More details are reported in Alg. 3 of the appendix.

## 4 EXPERIMENTAL EVALUATIONS

**Implementation Details** We implement our network in PyTorch and use the ADAM optimizer (Kingma & Ba, 2014) with a learning rate of 0.001. The point-transformation mapping network is implemented by two FC-layers composed of 64 hidden units. We set the initial activation of the input LRF to 1.0. In each layer, we use 3 iterations of DR. For classification we use the spread loss (Sabour et al., 2018) and the rotation loss is identical to $\delta(\cdot)$.

Surface normals are computed by local plane fits (Hoppe et al., 1992). We compute the second axis of the LRF, $\boldsymbol{\partial}_2$, by FLARE (Petrelli & Di Stefano, 2012), that uses the normalized projection of the point within the periphery of the support showing the largest distance, onto the tangent plane of the center: $\boldsymbol{\partial}_2 = \frac{\mathbf{P}_{\max} - \mathbf{P}}{\|\mathbf{P}_{\max} - \mathbf{P}\|}$. Note that using other LRFs such as SHOT (Tombari et al., 2010) or the more modern GFrames of Melzi et al. (2019) is possible. We found FLARE to be sufficient for our experiments. Prior to all operations, we flip all the LRF quaternions such that they lie on the northern hemisphere : $\{\mathbf{q}_i \in \mathbb{S}^3 : q_i^w > 0\}$.

Table 2: Error of rotation estimation in different categories of ModelNet10. Right side of the table denotes the objects with rotational symmetry, which we include for completeness. PCA-S refers to running PCA only on a resampled instance, while PCA-SR applies both rotations and resampling.

| Method | Avg. | No_Sym | Chair | Bed | Sofa | Toilet | Monitor | Table | Desk | Dresser | NS | Bathtub |
|---|---|---|---|---|---|---|---|---|---|---|---|---|
| Mean LRF | 0.41 | 0.35 | 0.32 | 0.36 | 0.34 | 0.41 | 0.34 | 0.45 | 0.60 | 0.50 | 0.46 | 0.32 |
| PCA-S | 0.40 | 0.42 | 0.60 | 0.53 | 0.46 | 0.32 | 0.12 | 0.47 | **0.23** | **0.33** | 0.43 | 0.55 |
| PCA-SR | 0.67 | 0.67 | 0.69 | 0.70 | 0.67 | 0.68 | 0.61 | 0.67 | 0.67 | 0.67 | 0.66 | 0.70 |
| PointNetLK | 0.37 | 0.38 | 0.43 | 0.31 | 0.40 | 0.40 | 0.31 | 0.40 | 0.33 | 0.39 | 0.38 | 0.34 |
| IT-net | 0.27 | 0.19 | 0.10 | 0.22 | 0.17 | 0.20 | 0.28 | **0.31** | 0.41 | 0.44 | 0.40 | 0.39 |
| Ours | 0.27 | 0.17 | 0.11 | 0.20 | 0.16 | 0.18 | 0.19 | 0.43 | 0.40 | 0.48 | 0.33 | 0.31 |
| Ours (siamese) | **0.20** | **0.09** | **0.08** | **0.10** | **0.08** | **0.11** | **0.08** | 0.40 | 0.35 | 0.34 | **0.32** | **0.30** |

**3D Shape Classification.** We use ModelNet40 dataset of (Wu et al., 2015; Qi et al., 2017b) to assess our classification performance. Each shape is composed by $10K$ points. We assign the LRFs to a subset of the uniformly sampled points, $N = 512$ (Birdal & Ilic, 2017). We train the networks without any rotation augmentation (NR) and put them to test under arbitrary $SO(3)$ rotations (AR). Our results are shown in Tab. 1 along with that of PointNet (PN) (Qi et al., 2017a), PointNet++ (PN++) (Qi et al., 2017a), KD-treeNet (Li et al., 2018a), Point2Seq (Liu et al., 2019b), Spherical CNNs (Esteves et al., 2018), PRIN (You et al., 2018) and PPF-FoldNet (PPF) (Deng et al., 2018a). We also present a version of our algorithm (*Var*) that avoids the canonicalization within the QE-network. This is a non-equivariant network that we still train without data augmentation. While this version gets comparable results to the state of the art for the NR/NR case, it cannot handle random $SO(3)$ variations (AR). Note that PPF uses the point-pair-feature (Birdal & Ilic, 2015) encoding and hence creates invariant input representations. For the scenario of NR/AR, our equivariant version outperforms all the other methods, including equivariant spherical CNNs (Esteves et al., 2018) by a significant gap of at least $5\%$ even when (Esteves et al., 2018) uses the mesh. The object rotational symmetries in this dataset are responsible for a significant portion of the errors we make. It is worth mentioning that we also trained TFNs (Thomas et al., 2018) for that task, but their memory demand made it infeasible to scale to this application.

**Number of Parameters.** Use of LRFs helps us to restrict the rotation group to certain elements and thus we can use networks with significantly less parameters (as low as $0.44M$) compared to others as shown in Tab. 1. Number of parameters in our network depends upon the number of classes, e.g. for ModelNet10 we have $0.047M$ parameters.

**Rotation estimation in 3D point clouds.** Our network can estimate both the canonical and relative object rotations without pose-supervision. To evaluate this desired property, we used the well classified shapes on ModelNet10 dataset, a sub-dataset of Modelnet40 (Wu et al., 2015). We generate multiple instances per shape by transforming the instance with five arbitrary $SO(3)$ rotations. As we are also affected by the sampling of the point cloud, we resample the mesh five times and generate different pooling graphs across all the instances of the same shape. Our QE-architecture can estimate the pose in two ways: 1) by directly using the output capsule with the highest activation, 2) by a siamese architecture that computes the relative quaternion between the capsules that are maximally activated as shown in Fig. 4. Both modes of operation are free of the data augmentation and we give further schematics of the latter in our appendix Fig. 5. Our results against the baselines including a naive averaging of the LRFs (Mean LRF) and principal axis alignment (PCA) are reported in Tab. 2 as the relative angular error (RAE). We further include results of PointNetLK Aoki et al. (2019) and IT-Net (Yuan et al., 2018), two state of the art 3D networks that iteratively aligns two point sets. These methods are in nature similar to iterative closest point (ICP) algorithm (Besl & McKay, 1992) but 1) do not require an initialization (first iteration estimates the pose), 2) learn data driven updates. Methods that use mesh inputs such as Spherical CNNs (Esteves et al., 2018) cannot be included here as the random sampling of the same surface would not affect those. We also avoid methods that are just invariant to rotations (and hence cannot estimate the pose) such as Tensorfield Networks (Thomas et al., 2018). Finally, note that , IT-net (Yuan et al., 2018) and PointLK need to train a lot of epoches (e.g. 500) with random $SO(3)$ rotation augmentation in order to get the models that cover the full $SO(3)$, whereas we train only for $\sim 100$ epochs. We include more details about the baselines in the appendix under Fig. 8.

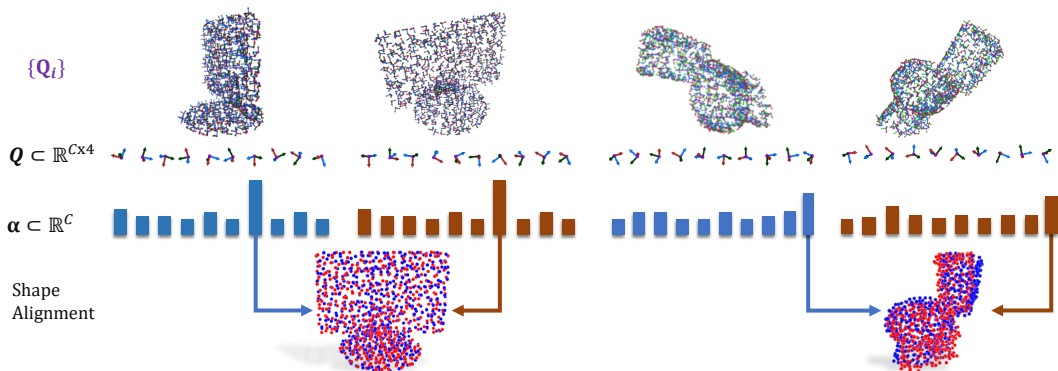

Figure 4: Shape alignment on the **monitor** (left) and **toilet** (right) objects via our siamese equivariant capsule architecture. The shapes are assigned to the the maximally activated class. The corresponding pose capsule provides the rotation estimate.

RAE between the ground truth and the prediction is computed as the relative angle in degrees: $d(\mathbf{q}_1, \mathbf{q}_2)/\pi$. Note that resampling and random rotations render the job of all methods difficult. However, both our version that tries to find a canonical alignment and the siamese variant which seeks a relative rotation are better than the baselines. As pose estimation of objects with rotational symmetry is a challenging task we also report results on the non-symmetric subset (No_Sym).

**Robustness against point and LRF resampling.** Density changes in the local neighborhoods of the shape are an important cause of error for our network. Hence, we ablate by applying random resamplings (patch-wise dropout) to the objects in ModelNet10 dataset and repeating the pose estima-

Table 3: Ablation study on point density.

| LRF Input | LRF-10K | | | | LRF-2K | LRF-1K |
|---|---|---|---|---|---|---|
| Dropout | 50% | 66% | 75% | 100% | 100% | 100% |
| Class. Err | 77.8 | 83.3 | 83.4 | 87.8 | 85.46 | 79.74 |
| Angle. Err | 0.34 | 0.27 | 0.25 | 0.09 | 0.10 | 0.12 |

tion and classification as described above. The first part (LRF-10K) of Tab. 3 shows our findings against gradual increases of the number of patches. Here, we sample 2K LRFs from the 10K LRFs computed on an input point 10K. 100% dropout corresponds to 2K points in all columns. On second ablation, we reduce the amount of points on which we compute the LRFs, to 2K and 1K respectively. As we can see from the table, our network is robust towards the changes in the LRFs as well as the density of the points.

## 5 RELATED WORK

**Deep learning on point sets.** The capability to process raw, unordered point clouds within a neural network is introduced by the prosperous PointNet (Qi et al., 2017a) thanks to the point-wise convolutions and the permutation invariant pooling functions. Many works have extended Point-Net primarily to increase the local receptive field size (Qi et al., 2017b; Li et al., 2018b; Shen et al., 2018; Wang et al., 2019). Point-clouds are generally thought of as sets. This makes any permutation-invariant network that can operate on sets an amenable choice for processing points (Zaheer et al., 2017; Rezatofighi et al., 2017). Unfortunately, common neural network operators in this category are solely equivariant to permutations and translations but to no other groups.

**Equivariance in Neural Networks.** The early attempts to achieve invariant data representations usually involved data augmentation techniques to accomplish tolerance to input transformations (Maturana & Scherer, 2015; Qi et al., 2016; 2017a). Motivated by the difficulty associated with augmentation efforts and acknowledging the importance of theoretically equivariant or invariant representations, the recent years have witnessed a leap in theory and practice of equivariant neural networks (Bao & Song, 2019; Kondor & Trivedi, 2018).

While laying out the fundamentals of the group convolution, G-CNNs (Cohen & Welling, 2016) guaranteed equivariance with respect to finite symmetry groups. Similarly, Steerable CNNs (Cohen & Welling, 2017) and its extension to 3D voxels (Worrall & Brostow, 2018) considered discrete symmetries only. Other works opted for designing filters as a linear combination of harmonic basis functions, leading to frequency domain filters (Worrall et al., 2017; Weiler et al., 2018b). Apart from suffering from the dense coverage of the group using group convolution, filters living in the

frequency space are less interpretable and less expressive than their spatial counterparts, as the basis does not span the full space of spatial filters.

Achieving equivariance in 3D is possible by simply generalizing the ideas of the 2D domain to 3D by voxelizing 3D data. However, methods using dense grids (Chakraborty et al., 2018; Cohen & Welling, 2017) suffer from increased storage costs, eventually rendering the implementations infeasible. An extensive line of work generalizes the harmonic basis filters to $SO(3)$ by using *e.g.*, a spherical harmonic basis instead of circular harmonics (Cohen et al., 2018b; Esteves et al., 2018; Cruz-Mota et al., 2012). In addition to the same downsides as their 2D, these approaches have in common that they require their input to be projected to the unit sphere (Jiang et al., 2019), which poses additional problems for unstructured point clouds. A related line of research are methods which define a regular structure on the sphere to propose equivariant convolution operators (Liu et al., 2019a; Boomsma & Frellsen, 2017)

To learn a rotation equivariant representation of a 3D shape, one can either act on the input data or on the network. In the former case, one either presents augmented data to the network (Qi et al., 2017a; Maturana & Scherer, 2015) or ensures rotation-invariance in the input (Deng et al., 2018a;b; Khoury et al., 2017). In the latter case one can enforce equivariance in the bottleneck so as to achieve an invariant latent representation of the input (Mehr et al., 2018; Thomas et al., 2018; Spezialetti et al., 2019). Further, equivariant networks for discrete sets of views (Esteves et al., 2019b) and cross-domain views (Esteves et al., 2019a) have been proposed. Here, we aim for a different way of embedding equivariance in the network by means of an explicit latent rotation parametrization in addition to the invariant feature.

Marcos et al. (2017) developed *Vector Field Networks*, which was followed by the 3D *Tensor Field Networks* (TFN) (Thomas et al., 2018) that are closest to our work. Based upon a geometric algebra framework, the authors did achieve localized filters that are equivariant to rotations, translations and permutations. Moreover, they are able to cover the continuous groups. However, TFN are designed for physics applications, is memory consuming and a typical implementation is neither likely to handle the datasets we consider nor can provide orientations in an explicit manner.

**Capsule Networks.** The idea of capsule networks was first mentioned by Hinton et al. (2011), before Sabour et al. (2017) proposed the *dynamic routing by agreement*, which started the recent line of work investigating the topic. Since then, routing by agreement has been connected to several well-known concepts, e.g. the EM algorithm Sabour et al. (2018), clustering with KL divergence regularization Wang & Liu (2018) and equivariance (Lenssen et al., 2018). They have been extended to autoencoders (Kosiorek et al., 2019) and GANs Jaiswal et al. (2019). Further, capsule networks have been applied for specific kinds of input data, e.g. graphs (Xinyi & Chen, 2019), 3D point clouds (Zhao et al., 2019) or medical images (Afshar et al., 2018).

## 6 CONCLUSION AND DISCUSSION

In this work, we have presented a new framework for achieving permutation invariant and $SO(3)$ equivariant representations on 3D point clouds. Proposing a variant of the capsule networks, we operate on a sparse set of rotations specified by the input LRFs thereby circumventing the effort to cover the entire $SO(3)$. Our network natively consumes a compact representation of the group of 3D rotations - quaternions, and we have theoretically shown its equivariance. We have also established convergence results for our Weiszfeld dynamic routing by making connections to the literature of robust optimization. Our network is among the few for having an explicit group-valued latent space and thus naturally estimates the orientation of the input shape, even without a supervision signal.

**Limitations.** In the current form our performance is severely affected by the shape symmetries. The length of the activation vector depends on the number of classes and for achieving sufficiently descriptive latent vectors we need to have a significant number of classes. On the other side, this allows us to perform with merit on problems where the number of classes are large. Although, we have reported robustness to those, the computation of LRFs are still sensitive to the point density changes and resampling. LRFs themselves are also ambiguous and sometimes non-unique.

**Future work.** Inspired by Cohen et al. (2019) and Poulenard & Ovsjanikov (2018) our feature work will involve establishing invariance to the direction in the tangent plane. We also plan to apply our network in the broader context of 3D object detection under arbitrary rotations and look for equivariances among point resampling.

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

## A PROOF OF PROPOSITION 1

Before presenting the proof we recall the three individual statements contained in Prop. 1:

1. $\mathcal{A}(\mathbf{g} \circ \mathbf{S}, \mathbf{w})$ is left-equivariant: $\mathcal{A}(\mathbf{g} \circ \mathbf{S}, \mathbf{w}) = \mathbf{g} \circ \mathcal{A}(\mathbf{S}, \mathbf{w})$.
2. Operator $\mathcal{A}$ is invariant under permutations: $\mathcal{A}(\{\mathbf{q}_{\sigma(1)}, \ldots, \mathbf{q}_{\sigma(Q)}\}, \mathbf{w}_\sigma) = \mathcal{A}(\{\mathbf{q}_1, \ldots, \mathbf{q}_Q\}, \mathbf{w})$.
3. The transformations $\mathbf{g} \in \mathbb{H}_1$ preserve the geodesic distance $\delta(\cdot)$.

*Proof.* We will prove the propositions in order.

1. We start by transforming each element and replace $\mathbf{q}_i$ by $(\mathbf{g} \circ \mathbf{q}_i)$ of the cost in Eq (4):

$$\mathbf{q}^\top \mathbf{M} \mathbf{q} = \mathbf{q}^\top \Big( \sum_{i=1}^{Q} w_i \mathbf{q}_i \mathbf{q}_i^\top \Big) \mathbf{q} \tag{5}$$

$$= \mathbf{q}^\top \Big( \sum_{i=1}^{Q} w_i (\mathbf{g} \circ \mathbf{q}_i)(\mathbf{g} \circ \mathbf{q}_i)^\top \Big) \mathbf{q} \tag{6}$$

$$= \mathbf{q}^\top \Big( \sum_{i=1}^{Q} w_i \mathbf{G} \mathbf{q}_i \mathbf{q}_i^\top \mathbf{G}^\top \Big) \mathbf{q} \tag{7}$$

$$= \mathbf{q}^\top \Big( \mathbf{G} \mathbf{M}_1 \mathbf{G}^\top + \cdots + \mathbf{G} \mathbf{M}_Q \mathbf{G}^\top \Big) \mathbf{q}$$

$$= \mathbf{q}^\top \mathbf{G} \Big( \mathbf{M}_1 \mathbf{G}^\top + \cdots + \mathbf{M}_Q \mathbf{G}^\top \Big) \mathbf{q} \tag{8}$$

$$= \mathbf{q}^\top \mathbf{G} \Big( \mathbf{M}_1 + \cdots + \mathbf{M}_Q \Big) \mathbf{G}^\top \mathbf{q} \tag{9}$$

$$= \mathbf{q}^\top \mathbf{G} \mathbf{M} \mathbf{G}^\top \mathbf{q} \tag{10}$$

$$= \mathbf{p}^\top \mathbf{M} \mathbf{p}, \tag{11}$$

where $\mathbf{M}_i = w_i \mathbf{q}_i \mathbf{q}_i^\top$ and $\mathbf{p} = \mathbf{G}^\top \mathbf{q}$. From orthogonallity of $\mathbf{G}$ it follows $\mathbf{p} = \mathbf{G}^{-1} \mathbf{q} \implies \mathbf{g} \circ \mathbf{p} = \mathbf{q}$ and hence $\mathbf{g} \circ \mathcal{A}(\mathbf{S}, \mathbf{w}) = \mathcal{A}(\mathbf{g} \circ \mathbf{S}, \mathbf{w})$.

2. The proof follows trivially from the permutation invariance of the symmetric summation operator over the outer products in Eq (8).

3. It is sufficient to show that $|\mathbf{q}_1^\top \mathbf{q}_2| = |(\mathbf{g} \circ \mathbf{q}_1)^\top (\mathbf{g} \circ \mathbf{q}_2)|$ for any $\mathbf{g} \in \mathbb{H}_1$:

$$|(\mathbf{g} \circ \mathbf{q}_1)^\top (\mathbf{g} \circ \mathbf{q}_2)| = |\mathbf{q}_1^\top \mathbf{G}^\top \mathbf{G} \mathbf{q}_2| \tag{12}$$

$$= |\mathbf{q}_1^\top \mathbf{I} \mathbf{q}_2| \tag{13}$$

$$= |\mathbf{q}_1^\top \mathbf{q}_2|, \tag{14}$$

where $\mathbf{g} \circ \mathbf{q} \equiv \mathbf{G} \mathbf{q}$. The result is a direct consequence of the orthonormality of $\mathbf{G}$.

$\square$

## B PROOF OF LEMMA 1

We will begin by recalling some preliminary definitions and results that aid us to construct the connection between the dynamic routing and the Weiszfeld algorithm.

**Definition 9** (Affine Subspace). *A $d$-dimensional affine subspace of $R^N$ is obtained by a translation of a $d$-dimensional linear subspace $V \subset \mathbb{R}^N$ such that the origin is included in $S$:*

$$S = \Big\{ \sum_{i=1}^{d+1} \alpha_i \mathbf{x}_i \mid \sum_{i=1}^{d+1} \alpha_i = 1 \Big\}. \tag{15}$$

*Simplest choices for $S$ involve points, lines and planes of the Euclidean space.*

**Definition 10** (Orthogonal Projection onto an Affine Subspace). *An orthogonal projection of a point* $\mathbf{x} \in \mathbb{R}^N$ *onto an affine subspace explained by the pair* $(\mathbf{A}, \mathbf{c})$ *is defined as:*

$$\Pi_i(\mathbf{x}) \triangleq proj_S(\mathbf{x}) = \mathbf{c} + \mathbf{A}(\mathbf{x} - \mathbf{c}). \tag{16}$$

$\mathbf{c}$ *denotes the translation to make origin inclusive and* $\mathbf{A}$ *is a projection matrix typically defined via the orthonormal bases of the subspace.*

**Definition 11** (Distance to Affine Subspaces). *Distance from a given point* $\mathbf{x}$ *to a set of affine subspaces* $\{S_1, S_2 \ldots S_k\}$ *can be written as Aftab et al. (2015):*

$$C(\mathbf{x}) = \sum_{i=1}^{k} d(\mathbf{x}, S_i) = \sum_{i=1}^{k} \|\mathbf{x} - proj_{S_i}(\mathbf{x})\|^2. \tag{17}$$

**Lemma 2.** *Given that all the antipodal counterparts are mapped to the northern hemisphere, we will now think of the unit quaternion or versor as the unit normal of a four dimensional hyperplane* $h$, *passing through the origin:*

$$h_i(\mathbf{x}) = \mathbf{q}_i^\top \mathbf{x} + q_d := 0. \tag{18}$$

$q_d$ *is an added term to compensate for the shift. When* $q_d = 0$ *the origin is incident to the hyperplane. With this perspective, quaternion* $\mathbf{q}_i$ *forms an affine subspace with* $d = 4$, *for which the projection operator takes the form:*

$$proj_{S_i}(\mathbf{p}) = (\mathbf{I} - \mathbf{q}_i \mathbf{q}_i^\top)\mathbf{p} \tag{19}$$

*Proof.* We consider Eq (19) for the case where $\mathbf{c} = \mathbf{0}$ and $\mathbf{A} = (\mathbf{I} - \mathbf{q}\mathbf{q}^\top)$. The former follows from the fact that our subspaces by construction pass through the origin. Thus, we only need to show that the matrix $\mathbf{A} = \mathbf{I} - \mathbf{q}\mathbf{q}^\top$ is an orthogonal projection matrix onto the affine subspace spanned by $\mathbf{q}$. To this end, it is sufficient to validate that $\mathbf{A}$ is symmetric and idempotent: $\mathbf{A}^\top \mathbf{A} = \mathbf{A}\mathbf{A} = \mathbf{A}^2 = \mathbf{A}$. Note that by construction $\mathbf{q}^\top \mathbf{q}$ is a symmetric matrix and hence $\mathbf{A}$ itself. Using this property and the unit-ness of the quaternion, we arrive at the proof:

$$\mathbf{A}^\top \mathbf{A} = (\mathbf{I} - \mathbf{q}\mathbf{q}^\top)^\top (\mathbf{I} - \mathbf{q}\mathbf{q}^\top) \tag{20}$$

$$= (\mathbf{I} - \mathbf{q}\mathbf{q}^\top)(\mathbf{I} - \mathbf{q}\mathbf{q}^\top) \tag{21}$$

$$= \mathbf{I} - 2\mathbf{q}\mathbf{q}^\top + \mathbf{q}\mathbf{q}^\top \mathbf{q}\mathbf{q}^\top \tag{22}$$

$$= \mathbf{I} - 2\mathbf{q}\mathbf{q}^\top + \mathbf{q}\mathbf{q}^\top \tag{23}$$

$$= \mathbf{I} - \mathbf{q}\mathbf{q}^\top \triangleq \mathbf{A} \tag{24}$$

It is easy to verify that the projections are orthogonal to the quaternion that defines the subspace by showing $proj_S(\mathbf{q})^\top \mathbf{q} = 0$:

$$\mathbf{q}^\top proj_S(\mathbf{q}) = \mathbf{q}^\top \mathbf{A}\mathbf{q} = \mathbf{q}^\top (\mathbf{I} - \mathbf{q}\mathbf{q}^\top)\mathbf{q} = \mathbf{q}^\top (\mathbf{q} - \mathbf{q}\mathbf{q}^\top \mathbf{q}) = \mathbf{q}^\top (\mathbf{q} - \mathbf{q}) = 0. \tag{25}$$

$$\tag{26}$$

Also note that this choice corresponds to $\text{tr}(\mathbf{q}\mathbf{q}^\top) = \sum_{i=1}^{d+1} \alpha_i = 1$. □

**Lemma 3.** *The quaternion mean we suggest to use in the main paper Markley et al. (2007) is equivalent to the Euclidean Weiszfeld mean on the affine quaternion subspaces.*

*Proof.* We now recall and summarize the $L_q$-Weiszfeld Algorithm on affine subspaces Aftab et al. (2015), which minimizes a $q$-norm variant of the cost defined in Eq (17):

$$C_q(\mathbf{x}) = \sum_{i=1}^{k} d(\mathbf{x}, S_i) = \sum_{i=1}^{k} \|\mathbf{x} - proj_{S_i}(\mathbf{x})\|^q. \tag{27}$$

Defining $\mathbf{M}_i = \mathbf{I} - \mathbf{A}_i$, Alg. 2 summarizes the iterative procedure.

Note that when $q = 2$, the algorithm reduces to the computation of a non-weighted mean ($w_i = 1 \, \forall i$), and a closed form solution exists for Eq (29) and is given by the normal equations:

$$\mathbf{x} = \left( \sum_{i=1}^{k} w_i \mathbf{M}_i \right)^{-1} \left( \sum_{i=1}^{k} w_i \mathbf{M}_i \mathbf{c}_i \right) \tag{30}$$

---

**Algorithm 2:** $L_q$ Weiszfeld Algorithm on Affine Subspaces Aftab et al. (2015).

---

1  **input:** An initial guess $\mathbf{x}_0$ that does not lie any of the subspaces $\{S_i\}$, Projection operators $\Pi_i$, the norm parameter $q$

2  $\mathbf{x}^t \leftarrow \mathbf{x}_0$

3  **while** *not converged* **do**

4     Compute the weights $\mathbf{w}^t = \{w_i^t\}$:

$$w_i^t = \|\mathbf{M}_i(\mathbf{x}^t - \mathbf{c}_i)\|^{q-2} \quad \forall i = 1\ldots k \tag{28}$$

5     Solve:

$$\mathbf{x}^{t+1} = \arg\min_{\mathbf{x}\in\mathbb{R}^N} \sum_{i=1}^{k} w_i^t \|\mathbf{M}_i(\mathbf{x} - \mathbf{c}_i)\|^2 \tag{29}$$

---

For the case of our quaternionic subspaces $\mathbf{c} = \mathbf{0}$ and we seek the solution that satisfies:

$$\Big(\sum_{i=1}^{k}\mathbf{M}_i\Big)\mathbf{x} = \Big(\frac{1}{k}\sum_{i=1}^{k}\mathbf{M}_i\Big)\mathbf{x} = \mathbf{0}. \tag{31}$$

It is well known that the solution to this equation under the constraint $\|\mathbf{x}\| = 1$ lies in nullspace of $\mathbf{M} = \frac{1}{k}\sum_{i=1}^{k}\mathbf{M}_i$ and can be obtained by taking the singular vector of $\mathbf{M}$ that corresponds to the largest singular value. Since $\mathbf{M}_i$ is idempotent, the same result can also be obtained through the eigendecomposition:

$$\mathbf{q}^\star = \arg\max_{\mathbf{q}\in\mathcal{S}^3} \mathbf{q}\mathbf{M}\mathbf{q} \tag{32}$$

which gives us the unweighted Quaternion mean Markley et al. (2007). $\qquad\qquad\square$

## C   Proof of Theorem 1

Once the Lemma 1 is proven, we only need to apply the direct convergence results from the literature. Consider a set of points $\mathbf{Y} = \{\mathbf{y}_1 \ldots \mathbf{y}_K\}$ where $K > 2$ and $\mathbf{y}_i \in \mathbb{H}_1$. Due to the compactness, we can speak of a ball $\mathcal{B}(\mathbf{o}, \rho)$ encapsulating all $\mathbf{y}_i$. We also define the $\mathcal{D} = \{\mathbf{x} \in \mathbb{H}_1 \mid C_q(\mathbf{x}) < C_q(\mathbf{o})\}$, the region where the loss decreases.

We first state the assumptions that permit our theoretical result. These assumptions are required by the works that establish the convergence of such Weiszfeld algorithms Luenberger et al. (1984); Aftab & Hartley (2015); Aftab et al. (2014) :

**H1.** $\mathbf{y}_1 \ldots \mathbf{y}_K$ should not lie on a single geodesic of the quaternion manifold.
**H2.** $\mathcal{D}$ is bounded and compact. The topological structure of $SO(3)$ imposes a bounded convexity radius of $\rho < \pi/2$.
**H3.** The minimizer in Eq (29) is continuous.
**H4.** The weighting function $\sigma(\cdot)$ is concave and differentiable.
**H5.** Initial quaternion (in our network chosen randomly) does not belong to any of the subspaces.

Note that **H5** is not a strict requirement as there are multiple ways to circumvent (simplest being a re-initialization). Under these assumptions, the sequence produced by Eq (29) will converge to a critical point unless $\mathbf{x}^t = \mathbf{y}_i$ for any $t$ and $i$ Aftab et al. (2014). For $q = 1$, this critical point is on one of the subspaces specified in Eq (18) and thus is a geometric median. $\qquad\square$

Note that due to the assumption **H2**, we cannot converge from any given point. For randomly initialized networks this is indeed a problem and does not guarantee practical convergence. Yet, in our experiments we have not observed any issue with the convergence of our dynamic routing. As our result is one of the few ones related to the analysis of DR, we still find this to be an important first step.

For different choices of $q : 1 \leq q \leq 2$, the weights take different forms. In fact, this IRLS type of algorithm is shown to converge for a larger class of weighting choices as long as the aforementioned conditions are met. That is why in practice we use a simple sigmoid function.

## D    OUR SIAMESE ARCHITECTURE AND THE ALGORITHM

For estimation of the relative pose with supervision, we benefit from a Siamese variation of our network. In this case, latent capsule representations of two point sets $\mathbf{X}$ and $\mathbf{Y}$ jointly contribute to the pose regression as shown in Fig. 5.

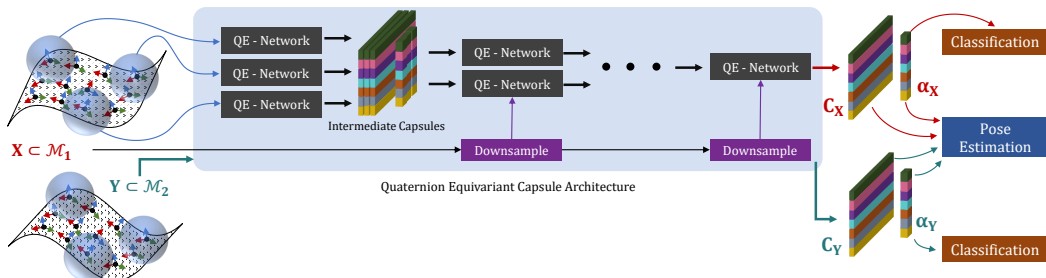

Figure 5: Our siamese architecture used in the estimation of relative poses. We use a shared network to process two distinct point clouds $(\mathbf{X}, \mathbf{Y})$ to arrive at the latent representations $(\mathbf{C}_X, \boldsymbol{\alpha}_X)$ and $(\mathbf{C}_Y, \boldsymbol{\alpha}_Y)$ respectively. We then look for the highest activated capsules in both point sets and compute the rotation from the corresponding capsules. Thanks to the rotations disentangled into capsules, this final step simplifies to a relative quaternion calculation.

We show additional results from the computation of local reference frames and the multi-channel capsules deduced from our network in Fig. 6.

Finally, the overall algorithm of our network is summarized under Alg. 3.

---

**Algorithm 3:** Quaternion Equivariant Network

---

1 **input  :** Input points of one patch $\{\mathbf{x}_1, ..., \mathbf{x}_K\} \in \mathbb{R}^{K \times 3}$, input capsules (LRFs)
   $\mathcal{Q} = \{\mathbf{q}_1, \dots, \mathbf{q}_L\} \in \mathbb{H}_1{}^L$, with $L = N^c \cdot K$, $N^c$ is the number of capsules per point, activations $\boldsymbol{\alpha} = (\alpha_1, \dots, \alpha_L)^T$
2 **output:** Updated frames $\hat{\mathcal{Q}} = \{\hat{\mathbf{q}}_1, \dots, \hat{\mathbf{q}}_M\} \in \mathbb{H}_1{}^M$, updated activations $\hat{\boldsymbol{\alpha}} = (\hat{\alpha}_1, \dots, \hat{\alpha}_M)^T$
3 **for** *Each input channel $n^c$ of all the primary capsules channels $N^c$* **do**
4 $\quad$ $\mu(n^c) \leftarrow \mathcal{A}(\mathcal{Q}(n^c))$ // Input quaternion average, see Eq (4)
5 $\quad$ **for** *Each point $\mathbf{x}_i$ of this patch* **do**
6 $\quad\quad$ $\mathbf{x}'_i \leftarrow \mu(n^c)^{-1} \circ \mathbf{x}_i$ // rotate point in a canonical orientation
7 $\{\mathbf{x}'_i\} \in \mathbb{R}^{K \times N^c \times 3}$ // Points in multiple $(N^c)$ canonical frames
8 **for** *Each point $\mathbf{x}'_i$ of this patch* **do**
9 $\quad$ $\mathbf{t} \leftarrow t(\mathbf{x}'_i)$ // Point to Transform, $t(\cdot) : \mathbb{R}^{N^c \times 3} \to \mathbb{R}^{N^c \times M \times 4}$
10 $\mathcal{T} \equiv \{\mathbf{t}_i\} \in \mathbb{H}_1{}^{K \times N^c_i \times M} \leftarrow \{\mathbf{t}\} \in \mathbb{H}_1{}^{L \times M}$
11 $(\hat{\mathcal{Q}}, \hat{\boldsymbol{\alpha}}) \leftarrow \text{DynamicRouting}(X, \mathcal{Q}, \boldsymbol{\alpha}, \mathcal{T})$ // see Alg. 1

---

Alg. 3 summarizes the overall pipeline of our QE-net depicted in Fig. 3. We use multiple layers in a hierarchical architecture. In the first layer, the input primary capsules are represented by LRFs computed with FLARE algorithm Petrelli & Di Stefano (2012). Therefore, the number of input capsule channels $N^c$ in the first layer is equal to $1$. Its activation is also defaulted to $1$. The output of a former layer is propagated to the input of the latter, creating the hierarchy.

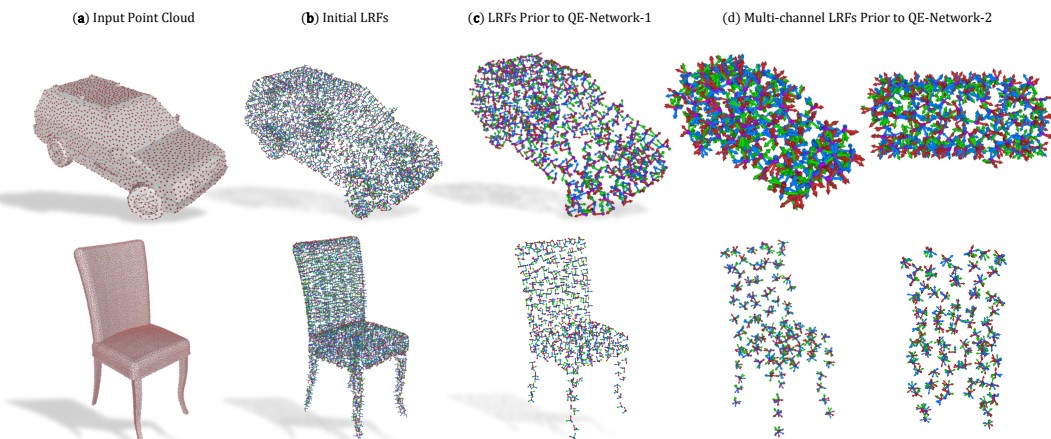

Figure 6: Additional intermediate results on car (first row) and chair (second row) objects. This figure supplements Fig. 1 of the main paper.

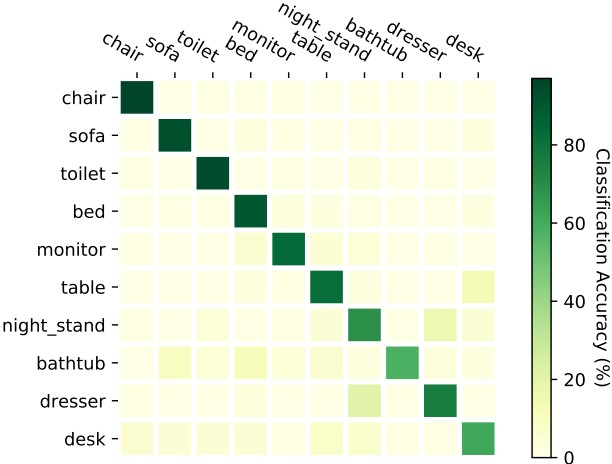

Figure 7: Confusion matrix on ModelNet10 for classification.

## E ADDITIONAL DETAILS ON EVALUATIONS

**Details on the evaluation protocol.** For Modelnet40 dataset used in Tab. 1, we used the official split with 9,843 shapes for training and 2,468 different shapes for testing. For rotation estimation in Tab. 2, we used the official Modelenet10 dataset split with 3991 for training and 908 shapes for testing. 3D point clouds (10K points) are randomly sampled from the mesh surfaces of each shape Qi et al. (2017a;b). The objects in training and testing dataset are different, but they are from the same categories so that they can be oriented meaningfully. During training, we did not augment the dataset with random rotations. All the shapes are trained with single orientation (well-aligned). We call this *trained with NR*. During testing, we randomly generate multiple arbitrary $SO(3)$ rotations for each shape and evaluate the average performance for all the rotations. This is called *test with AR*. This protocol is used in both our algorithms and the baselines.

**Confusion of classification in ModelNet.** We now report the confusion matrix in the task of classification on the all the objects of ModelNet10. The classification and rotation estimation affects one another. As we can see from Fig. 7, the first five categories that exhibit less rotational symmetry has the higher classification accuracy than their rotationally symmetric counterparts.

**Distribution of errors reported in Tab. 2.**    We now provide more details on the errors attained by our algorithm as well as the state of the art. To this end, we report, in Fig. 8 the histogram of errors that fall within quantized ranges of orientation errors. It is noticeable that our Siamese architecture behaves best in terms of estimating the objects rotation. For completeness, we also included the results of the variants presented in our ablation studies: Ours-2kLRF, Ours-1kLRF. They evaluate the model on the re-calculated LRFs in order to show the robustness towards to various point densities. We have also modified IT-Net and PointNetLK only to predict rotation because the original works predict both rotations and translations. Finally, note here that we do not use data augmentation for training our networks (see AR), while both for PointNetLK and for IT-Net we do use augmentation.

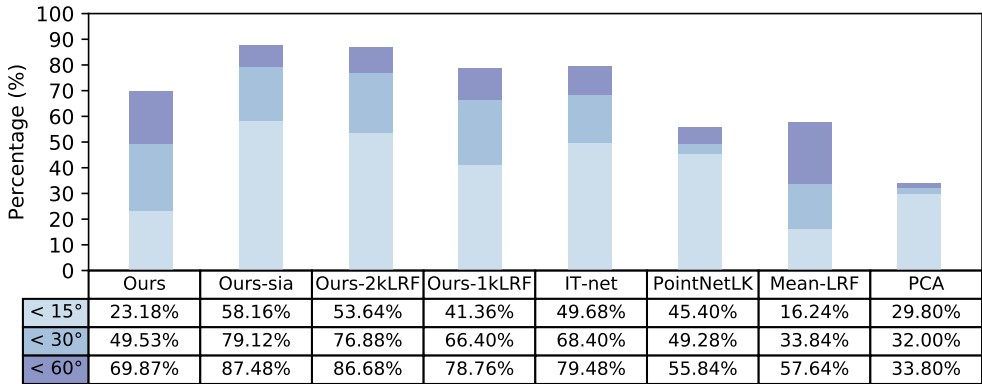

| | Ours | Ours-sia | Ours-2kLRF | Ours-1kLRF | IT-net | PointNetLK | Mean-LRF | PCA |
|---|---|---|---|---|---|---|---|---|
| < 15° | 23.18% | 58.16% | 53.64% | 41.36% | 49.68% | 45.40% | 16.24% | 29.80% |
| < 30° | 49.53% | 79.12% | 76.88% | 66.40% | 68.40% | 49.28% | 33.84% | 32.00% |
| < 60° | 69.87% | 87.48% | 86.68% | 78.76% | 79.48% | 55.84% | 57.64% | 33.80% |

Figure 8: Cumulative error histograms of rotation estimation on ModelNet10. Each row ($< \theta°$) of this extended table shows the percentage of shapes that have rotation error less than $\theta$. The colors of the bars correspond to the rows they reside in. The higher the errors are contained in the first bins (light blue) the better. Vice versa, the more the errors are clustered toward the $60°$ the worse the performance of the method.

