# OpenReview forum: "Quaternion Equivariant Capsule Networks for 3D Point Clouds"
_ICLR.cc/2020/Conference — Reject_

### Official Review · AnonReviewer3 · 2019-10-22
**Official Blind Review #3**

**Rating:** 3

**Review:**

This paper presents a rotation-equivariant model for use with point-clouds.  Capsule networks with activation and rotation pose estimate in the form of a quaternion are applied to local patches of 3D points, and combined up through progressively more coarser receptive fields.  Rotation equivariance is obtained by looking for agreement between input point frames (or capsules), by iteratively clustering using the quaternion mean.  The method is evaluated on ModelNet40, obtaining much improved results in the case where rotation orientation is arbitrary.

Unfortunately, I found too many important parts of the method difficult to understand, enumerated below, and am also not very clear on the details of how they fit together.  At a high level, the approach of aggregating pose agreement with quaternion averages makes intuitive sense, the quaternion average step itself is described well, and the experimental results seem to corroborate this (results on NR/AR condition are very good, and also near identical to NR/NR condition, a major verification of the method).  However, the rest of the pieces of the method still leave me guessing too much.

*  It isn't clear what computes the activations alpha.  I don't see a description for how to compute activations alpha that are the input to Alg. 1, unless these are simply the activations of the previous layer's capsules, and constant for the initial point frames?

*  Likewise, the transform t() could use more description.  Sec 3.2 says it is R3->R4, indicating that the input to t() is a single point.  Does this mean that the mere presence of a point in a location R3 (relative to a mostly-canonicalized pose) counts as a vote for a rotation pose?  And if so, how does does this work?  Or does the regressor actually take multiple points as input?  According to the picture in Fig. 1, it looks like it takes all points in the local patch as input.

*  Algorithm 1 outer for loop for i doesn't seem quite right.  It suggests that the body is performed either independently or in sequence for each i.  I believe what may actually be intended, is that all i are used at once in parallel, so that the v_ij votes assignment is found for *all* i in order to use in finding the cluster mean for each j.  That is, "v_i,j = Q_i * t_i,j for all i", rather than having the outermost loop over i.  Is this correct?

*  Much space is dedicated to background explanations of quaternions, geodescic distance, etc.  Given my difficulty in understanding much of the system, it seems like some of could be shortened or put into the appendix, and more of the main text devoted to more detailed explanations of each component in the system (e.g. alpha and t), and how the capsule agreement step (QE DR) makes use of these.


Also, some more detailed comments/suggestions:

* Alg.1:  output alpha^hat_K, seems K should be M here?
* Table 1:  "Right hand side denotes symmetric objects" seems it actually shoudl be in the caption for Table 2.


**Experience Assessment:**

I have read many papers in this area.

**Review Assessment: Checking Correctness Of Derivations And Theory:**

I assessed the sensibility of the derivations and theory.

**Review Assessment: Checking Correctness Of Experiments:**

I assessed the sensibility of the experiments.

**Review Assessment: Thoroughness In Paper Reading:**

I read the paper at least twice and used my best judgement in assessing the paper.

---

> ### Author Response · Authors · 2019-11-13
> **Our response to Official Blind Review #3**
>
> We thank the reviewer for the specific comments and acknowledging the improved results we achieve in the paper. Please find our responses below.
>
> 1. “The activation”: It is true that the update of alpha is only included in the algorithm as it is a standard result from previous works [Sabour et al. 2017a, Lenssen et al., 2018]. The input to Alg 1. are the alphas of the previous layer and we set the initial activations to 1. We have now clearly indicated that in the last paragraph of Sec.3.2.
>
> 2. “The transform t()”: We have significantly increased our explanation and reworked the formalization of the transform network in the paper. The t() network does not directly compute the vote. It is merely producing the transformation, on which the input pose is applied to produce the vote: v = q t(). It is the continuous analogue to the approach in the original capsule networks by Sabour et al. There, those transformations lie in a discrete kernel window and are directly optimized like weights for convolution.  Here, we train a continuous kernel function instead, given that input capsules are attached to points that lie in continuous space and not on a fixed grid. The function t() is shared for all input points and produces transformations for all combinations of input capsules from the respective point and output capsules, which we clarified in the first paragraph of Sec.3.2. Additionally, we added specific details of the architecture with pseudocode in Alg.3.
>
> 3.”Loop in Alg.1”: Thank you for making us aware of this error. We fixed the algorithm in the updated version.
>
> 4.”Limited space for explanation”: We have now shortened the background section to give room for further explanations. The paper is also slightly longer. As mentioned above, we have added more explanation of the network architecture in Sec.3.2 (added alpha and t), Sec.4 and appendix(Alg.3). In Alg.3, we explain the details of the QE-net with pseudocode, included the “t(.)” and how the DR is used after we got the transformations.
>
> 5.” more detailed comments”:  Thanks for those corrections. We have now incorporated them.

---

### Official Review · AnonReviewer1 · 2019-10-24
**Official Blind Review #1**

**Rating:** 6

**Review:**

This paper proposes a quaternion equivariant network for 3d point clouds. It builds on the capsule networks and dynamic routing, but instead of using arbitrary 4x4 transforms, they propose to use quaternions to represent 3D rotations.

Invariances and equivariances are very important in DNN for classification/detection, so extending these properties to 3d point clouds based applications is interesting and important. The proposed approach is novel and seems to be equivariant to SO(3) rotations, translations, and set permutations.

However, there are a couple of major negatives in the paper. The first is that the details of the architecture are very unclear. A large amount of space is allocated to discussing equivariant properties but not enough is dedicated to the actual implementations of the network. In particular,  only on page 5 in section 3.2 is the QE network detailed discussed.

- How does one obtain Q_i from points X_i ?
- How are the points subsampled to create LRFs? is there a clustering first step? If so, how would that clustering be invariant to global rotation?
- Does the method require a known centroid of the object point clouds in order to compute the quaternions of the LRFs?
- Theorem 1 has shown that equation 6 of Alg. 1 is equivariant, but are the multiple interactions of DR equivariant? In particular, after k iterations of DR as presented in algorithm 1, with nonlinear sigmoid activations, are the output capsule representations equivariant?
- How do you guarantee that the LRF’s are the same under any global rotation+translations of the point cloud? Or do you assume the same LRFs are known?

The main selling point for equivarince and learning the separation of pose and object class representation is in the hope of more accurate classification. However, empirical classification results are far from the state-of-the-art.
- As a side question, there are other methods listed on the ModelNet40 website with performances in the high 90’s, are there reasons those results are not included as a part of Table 1?

**Experience Assessment:**

I have read many papers in this area.

**Review Assessment: Checking Correctness Of Derivations And Theory:**

I assessed the sensibility of the derivations and theory.

**Review Assessment: Checking Correctness Of Experiments:**

I assessed the sensibility of the experiments.

**Review Assessment: Thoroughness In Paper Reading:**

I read the paper at least twice and used my best judgement in assessing the paper.

---

> ### Author Response · Authors · 2019-11-13
> **Our response to Official Blind Review #1**
>
> We thank the reviewer for finding our problem important and our approach novel. Please find our responses  below.
>
> 1. “Limited space for architecture details”:  We have now slightly shortened the background section and rewrote the descriptions about the architecture, the t-network and QE-network in Sec.3.2. The specific details of the architecture are also presented by the pseudocode in Alg.3 in order to make the work easier to replicate. We also increased the implementation details in Sec. 4.
>
> 2. “How to obtain Q_i from X_i” : This is done by computing local reference frames. We have now mentioned this explicitly.
>
> 3. “How to create LRF with point sampling“: In section 4 we have indicated that we use uniformly sampled points and cite the related literature that we utilize. While our work has theoretical equivariance properties (we do not discretize), it is affected by the choice of sampling. This is the reason why we included the resampling (point density) experiment. The LRFs are calculated for each patch using a center point and its neighbors. This is done before we perform uniform sampling to get the pooling centers hierarchically. The uniform sampling is not rotation invariant and hence might result in different pooling centers and this, as we have mentioned, is the primary source of error in our network. We have evaluated how changes in sampling affect our network in the last paragraph of the experiments section (Table 3). Note that during training, we feed the network with multitudes of different samplings and as a result, the network is trained to deal with the variance caused by this kind of data discretization. We also added Fig.8 in the appendix to show the robustness of our proposed algorithm by computing the LRF with different point densities.
>
> 4. “Object centroid for LRF computation”: For computing the LRFs we do not need a known centroid. These computations are done only locally. However, to transform them into the same coordinate frame we assume a common origin. Note that this does not have to be the center of the object, even though we use that piece of information in the paper.
>
> 5. “Equivariance towards non-linearities”:  Note that all the non-linearities are captured in the weights. Our proof is independent of the weighting scheme and we prove equivariance for an arbitrary (up to the provided conditions) choice of real weights. Also chaining k-consecutive equivariant operators would still preserve equivariance. This is a standard result and we have mentioned this in Dfn. 2. Nevertheless, we have mentioned this again when we speak of the architecture.
>
> 6. “LRF after shape transformation”: There might be a misunderstanding here. LRFs do not need to be the same. In fact, they are expected to transform equivariantly, e.g. the surface normal of a point rotates with R when the object is rotated by R. We do not assume that LRFs are given to us. We compute them by FLARE as we have aptly cited. It is worth mentioning again here is that we now report more results in Fig.8 of the appendix with the LRFs calculated in different point cloud densities to show the robustness.
>
> 7. “Classification performance”: In the paper, we already reported one such work “Point2Seq” that achieves 92.6% performance. However, most of those methods are not truly equivariant to rotations. This is why they suffer under the NR/AR case. However, our equivariant network can classify shapes in random rotations even only trained with shapes of such categories in single orientation(aligned). Note that our contributions can also be applied to more complex networks. As we use PointNet-like layers, we believe that it is best to evaluate against the methods of a similar nature. On another aspect, note that we also estimate canonical object poses, and not only the pairwise ones.

---

### Official Review · AnonReviewer2 · 2019-10-24
**Official Blind Review #2**

**Rating:** 6

**Review:**

This work builds a capsule network, for use with point cloud data, that has units that are equivariant to SO(3) 3D rotations.  I found that the authors made the case for rotation equivariance well and I liked the analysis of the dynamic routing approach and its mapping to the Generalized Weiszfeld Iterations.

As a caveat to the review I should point out that while I am familiar with capsule networks, they are not my main area of expertise (I do work on geometry however) so this should be taken into consideration - apologies if there is literature I have missed.

My main concern is that group equivariance for Capsule networks has been studied before - I acknowledge that SO(3) can present additional challenges but would have expected a work that specifically targeted SO(3) to investigate the relative benefits of the different approaches to encoding the lie algebra and not just quaternions. How do we know that quaternions are the best fit for the Capsule framework? Could the authors show empirical results that indicate that other representations for the equivariance do not work as well? My experience from optimization is that the best representation can often be application specific and it would be very helpful to understand their relative merits for this sort of architecture.

The preparation of the local patches and the estimation of the rotations seems very important to these applications to me and would be worthy of more discussion and an empirical presentation of their different merits. I know the authors discussed the issue of the second axis but it would seem worth including an experiment to detect the sensitivity in a similar manner to the point cloud density.

My biggest concern about the paper is the presentation of the results. Please could the authors provide appropriate error bars for the tables - this is the least action necessary to at least start to estimate significance. Ideally something like histograms or violin plots would be more useful. The use of a single dataset makes it hard to identify the efficacy of the approach - especially when it is not clear how the training and test data are set-up. Are the same objects used in training and test but with different rotations? Or different objects from the same categories. What is the protocol for the AR test set?

I agree that table 1 confirms the equivariance so that is great but it is less clear that the network is suitable for the pose task since the baselines seem very simple for the pose estimation experiment (and the related work covers a lot of prior work in this area) and again, without error bars it is hard to judge significance.

It is nice that the authors discuss limitations and I agree with the comment on symmetries - would it be possible to include an illustrative example to confirm these suspicions?


Other notes:

I have no issue with the proofs - they all seemed to make sense to me.

I found Figure 1 (and to a lesser extent Figure 3) to be unclear and uninformative - the notation doesn't match the main part of the paper (also where does C come from in the caption?) and its not clear what the illustrations are meant to represent. Would it not be much more clear to include something like Algorithm 1 instead of Figure 1 which is much more clear and easy to replicate the precise process.

I think there is a lot of material in the appendix that should really be in the main paper - I'm happy for the proofs to be in the appendix but I think it is a bit wrong to essentially violate the page restrictions by moving important related work into the appendix. The whole of section 2 is standard text-book information about quaternions - would it not be more appropriate for that to be in the appendix and the related work to be in the main paper?

Could the authors indicate the use of the Frobenius norm in eqn (5)? There also seems to be a mixing of notations for inner products through-out the paper and it might be helpful to standardize this?

**Experience Assessment:**

I do not know much about this area.

**Review Assessment: Checking Correctness Of Derivations And Theory:**

I carefully checked the derivations and theory.

**Review Assessment: Checking Correctness Of Experiments:**

I assessed the sensibility of the experiments.

**Review Assessment: Thoroughness In Paper Reading:**

I read the paper thoroughly.

---

> ### Author Response · Authors · 2019-11-13
> **Our response to Official Blind Review #2**
>
> We thank the reviewer positive feedback, going through the theoretical analysis and constructive criticism. We have revised the manuscript according to her/his concerns.
>
> 1. “On the use of quaternions”: It is true that multiple representations for rotations do exist and seem potentially suitable for the task at hand. While we do not theoretically know whether the quaternions are indeed the “best” choice (or if any such choice ought to exist), we do have multiple strong motivations for why quaternions (being one of the long-standing standards) are one reasonable choice: 1. Single redundancy, 2. No singularities, 3. Closed form mean that is differentiable and batch-friendly (see source code). 4. As acknowledged by the reviewer, the employed mean also allows us to draw connections to existing IRLS methods (Weiszfeld) enabling us to provide theoretical guarantees.
>
> Thanks for bringing this important point up. We have now included this discussion in Sec. 2.2 of the main paper.
>
> 2. “Sensitivity to point density”: We agree that this is an important issue. We have now included an ablation study in the main paper that analyzes the sensitivity of the local reference frame we employed. The results are tabulated in Tab. 3, where we gradually decrease the number of points used in the LRF computation besides the point set size used in the actual network. This way, we show that even when the number of points is reduced to 2K and 1K respectively, we are still able to get acceptable results.
>
> 3. “Presentation of results”: We have now included error bars and more details of the classification result as suggested in Fig.6 and Fig, 7. ModelNet is a standard dataset that is used by many works to measure the performance. We have chosen that dataset for fairer comparison to the existing works. As others, we believe this is adequate. Although our experimental evaluation follows the standard protocols of Qi et. al. 2017, we now included more details on how we prepare the data in the appendix and main paper. To summarize:
>       1> "Training and test data set-up": We used the official split for training and testing.[Qi et. al. 2017b]
>       2> "Objects used in training and test?" : The objects in training and testing dataset are different, but they are from the same categories so that they can be oriented meaningfully.
>       3> "Protocol for the AR test":  During testing, we randomly generate arbitrary SO(3) rotations for each shape in the test dataset and do the classification evaluation.
>
> 4. “More on pose estimation”: The literature is mainly dominated by three types of works: 1. Using meshes/multiviews as input representations, which might not be available under noisy real scenarios, 2. Using equivariance to yield invariant representations but not explicitly evaluating the rotation errors (such as Tensorfield Networks), 3. Proof-of-concept equivariant networks that are quite challenging to run on the large datasets we have. Nevertheless, in order to address the concern of the reviewer, we have now included an additional evaluation against the recent work of Iterative Transformer networks (IT-Net) that can simultaneously classify and estimate the pose of an object. The results can be found in Table 2 as well as in Figure 6 of the appendix. Note that we do not use data augmentation during training. The relevant discussions are added to the manuscript.
>
> 5. “Regarding limitations”: Other than symmetries, we do not observe any particular pattern in the way the errors are made in our network at a first glance. We leave it as future work to thoroughly analyze the characteristics of the errors and hope that our evaluations are enough indicators of the performance of our network.
>
> 6. “Further notes”: We fixed C_i to {q_i} now and made the notation in the figure consistent with the text. The illustrations represent architecture. As our network is quite different from a typical one, we tried to clarify these in the text. We also moved the figure to page 4 where its details are given and included Figure 2 as the initial figure, which summarizes the idea of aggregating LRFs. We also double-checked the consistency of the figure, algorithm and text. Thanks for mentioning these. As suggested by the reviewer, we now included the algorithm corresponding to the QE-network in the appendix (Alg. 3). We also added more details in the architecture section(Section 3.2).
>
> We have now removed the definition of d_{riemann} as it was practically not used. We keep the geodesic distance expressed in terms of quaternions.  We now consistently use the circle operator to refer to the Hadamard product and no-operator implies a dot product. Thank you for bringing this up.

---

> > ### Comment · AnonReviewer2 · 2019-11-15
> > **Thanks**
> >
> > Thank you for your detailed comments and the responses all make sense to me - very happy to remain supportive of the paper.

---

### Decision · Program_Chairs · 2019-12-19

**Decision:**

Reject

**Comment:**

This paper presents a capsule network to handle 3d point clouds which is equivariant to SO(3) rotations. It also provides the theoretical analysis to connect the dynamic routing approach to the Generalized Weiszfeld Iterations. The equivariant property of the method is demonstrated on classification and orientation estimation tasks of 3D shapes.
While the technical contribution of the method is sound, the main concern raised by the reviewers was the lack of details in the presentation of methodology and results. Although the authors have made substantial efforts to update the paper, some reviewers were still not convinced and thus the scores remained the same. The paper was on the very borderline, but because of the limited capacity, I regret that I have to recommend rejection.
Invariances and equivariances are indeed important topics in representation learning, for which the capsule network is known as one of the promising approaches but still not well investigated compared to other standard architectures. I encourage authors to resubmit the paper taking in the reviewers' comments.